# Essential Oil Composition, Antioxidant Activity and Leaf Micromorphology of Five Tunisian *Eucalyptus* Species

**DOI:** 10.3390/antiox12040867

**Published:** 2023-04-03

**Authors:** Flavio Polito, Florinda Fratianni, Filomena Nazzaro, Ismail Amri, Habiba Kouki, Marwa Khammassi, Lamia Hamrouni, Paola Malaspina, Laura Cornara, Sana Khedhri, Benedetta Romano, Daniela Claudia Maresca, Angela Ianaro, Giuseppe Ercolano, Vincenzo De Feo

**Affiliations:** 1Department of Pharmacy, University of Salerno, Via Giovanni Paolo II, 132, 84084 Fisciano, Italy; 2Institute of Food Science, CNR-ISA, Via Roma, 64, 83100 Avellino, Italy; 3Laboratory of Biotechnology and Nuclear Technology, National Center of Nuclear Science and Technology, Sidi Thabet, B.P. 72, Ariana 2020, Tunisia; 4Department of Earth, Environment and Life Sciences, University of Genova, Corso Europa 26, 16132 Genova, Italy; 5Faculty of Science, Bizerte, Zarzouna 7021, Tunisia; 6Department of Pharmacy, School of Medicine and Surgery, University of Napoli Federico II, Via D. Montesano, 49, 80131 Napoli, Italy

**Keywords:** *Eucalyptus*, leaf anatomy, phytochemical profile, antioxidant tests, reactive oxygen species, anti-inflammatory activity

## Abstract

*Eucalyptus* species have been widely employed in the projects of reforestation in Tunisia. Although their ecological functions are controversial, these plants are indeed important to counteract soil erosion, and represent a fast-growing source of fuelwood and charcoal wood. In the present study, we considered five *Eucalyptus* species, namely *Eucalyptus alba*, *E. eugenioides*, *E. fasciculosa*, *E. robusta*, and *E. stoatei* cultivated in the Tunisian *Arboreta*. The aim was to carry out the micromorphological and anatomical characterization of the leaves, the extraction and phytochemical profile of the essential oils (EOs), and the evaluation of their biological properties. Four of the EOs showed the prevalence of eucalyptol (1,8-cineole) varying from 64.4 to 95.9%, whereas *a*-pinene predominated in *E. alba* EO (54.1%). These EOs showed in vitro antioxidant activity, and reduced the oxidative cellular stress as shown by their activity on reactive oxygen species (ROS) production, and modulation of the expression of antioxidant enzymes, such as glutamate-cysteine ligase (GCL) and heme oxygenase-1 (Hmox-1). Moreover, the EOs inhibited the production of nitric oxide (NO), showing anti-inflammatory activity. The data collected suggest that these EOs may be considered a promising therapeutic strategy for inflammation-based diseases and may represent an additional value for the economy of Tunisia.

## 1. Introduction

Genus *Eucalyptus* (Myrtaceae) includes about 800 species that are mostly endemic to Australia but actually distributed throughout the world and employed for many different applications [1]. The most important species globally used as source of EO with therapeutical properties is *Eucalyptus globulus* Labill [2]. *E. camaldulensis* Dehnh. was particularly quoted for its antimicrobial properties among *Eucalyptus* species traditionally used by aboriginal people of Australia. *E. grandis* W.Hill, *E. smithii* F.Muell. ex R.T.Baker, *E. nitens* (H.Deane & Maiden) Maiden, *E. dunni* Maiden, *E. globulus* and *E. urophylla* S.T.Balke are the most important species for obtaining the dissolving pulp that provides cellulose used in the textile and paper industries [3].

*Eucalyptus* has been widely used since the late 1950s in support of the major projects of reforestation in Tunisia, where more than one hundred species have been introduced, and therefore nowadays they can be found all over the country in different *Arboreta*, together with other introduced tree species belonging to the *Acacia*, *Pinus,* and *Casuarina* genera [4]. Although their ecological functions are controversial [5], Eucalyptus trees are important to counteract soil erosion [4]. In addition, due to their wide adaptability [6] and high productivity [7], they represent a fast-growing source of wood, which is then used for construction and furniture, and as firewood, generating large economic returns [8]. *Eucalyptus* are also melliferous species, and their important unifloral honey is highly requested by consumers for its health-promoting properties and marketed worldwide [9,10].

Among the *Eucalyptus* species introduced to Tunisia for the intensive reforestation, many different characteristics and utilities can be found, and this leads to the possibility of selecting the best-performing species for different specific applications, also considering the benefit that each of them can give to the general economy of the country. For example, *Eucalyptus astringens* (Maiden) Maiden, *E. maculata* (Hook.) K.D. Hill & L.A.S. Johnson and *E. robusta* Sm. offer good biomass production and good quality fuelwood and charcoal. *E. lehmanii* (Schauer) Benth. is a tree of bioenergetic interest but it is also a species of interest for honey production, similarly to *E. longifolia* Link, which can even be grown on highly saline and clayey soils [11]. Moreover, Eucalyptus EOs are widely employed as antimicrobial [12], antifungal [13], antiseptic [14], and disinfecting agents and for wound healing [15]. These EOs, due to their antimicrobial efficacy, are also widely used in cosmetic products, such as toothpastes, and mouthwashes [16].

A recent study reported the inhibitory effect on a bacterial biofilm of EOs from different Tunisian *Eucalyptus* species [17]. Moreover, it showed that the abundance of eucalyptol (synonym = 1,8-cineole) and terpinene derivatives, with phytotoxic properties, makes it possible to also use Eucalyptus EOs for agricultural application as natural pesticides [18,19]. In addition, the main component of Eucalyptus EOs, eucalyptol, has degreasing and solvent-dissolving properties, which increased the commercial potential of this product during the 1990s, following measures taken to phase out the petrochemical-based trichloroethane, a chemical that depletes the ozone layer [20].

EOs are also an interesting source of natural antioxidants, which can be used to replace synthetic antioxidants such as butylhydroxyanisole (BHA) and *tert*-butyl hydroquinone (TBHQ), which are seriously hazardous to human health [21].

EOs can be also directly added to edible products or used for packaging and edible coatings, as a valid method to prevent autoxidation and to improve products’ shelf life [22]. Moreover, due to their antioxidant effect, many EOs can be employed in the treatment of diseases with inflammatory aspects [23].

All these properties depend on the EO’s phytochemical profile, although it is difficult to find a clear correlation between the antioxidant potentials and the components, due to the chemical complexity of many EOs. Zhao and Coworkers [24] reported that the highly antioxidant properties of *E. citriodora* (Hook.) K.D. Hill & L.A.S. Johnson and *E. staigeriana* F. Muell. ex F.M. Bailey were likely due to the relatively high abundance of geranial, neral, citronellal, terpinolene and terpinene, which are rather active in scavenging ROS [25,26]. On the other hand, Siramon and Ohtani [27] reported the significant antioxidant and antiradical activities of the EO from *E. camaldulensis* growing in Thailand, mainly due to the presence of phenolic compounds like thymol and carvacrol.

Therefore, the aim of the present study was to analyze the EOs obtained from the leaves of five *Eucalyptus* species growing in Tunisia, which are scarcely investigated to date from this point of view. The study included the anatomical and micro-morphological characterization of the leaves, the extraction and phytochemical profile determination of the EOs, and the evaluation of their antioxidant and anti-inflammatory properties. The possibility of using the EOs obtained by the leaves of these species is interesting as they represent an important source of phytotherapeutical products. In addition, from a circular economy perspective, considering the large quantities of leafy branches disposed of as a result of cutting for industrial exploitation, these plant materials may represent additional value for the economy of Tunisia.

## 2. Materials and Methods

### 2.1. Plant Material

Leafy branches of five species of Eucalyptus trees, namely *Eucalyptus alba* Reinw. ex Blume, *E. eugenioides* Sieber ex Spreng., *E. fasciculosa* F. Muell., *E. robusta* Sm., and *E. stoatei* C. Gardner, were collected during October 2022. These species were introduced to Tunisia in 1954 and planted between the years 1959–1964 in Zernisa *Arboretum* (latitude 37°16′N; longitude 9°36′E; altitude 99 m), located in the region of Sejnane (Northern Tunisia). The trees were planted in separate plots at the equal distance of 6 m (width and length). The *Arboretum* is characterized by poorly developed soil in coastal dunes with leached brown forest in the mountains. The climate is subhumid with an annual rainfall of 927 mm. Mean annual temperatures ranged from 14.9 to 18.5 °C. The average minimum of the coldest month is around 4 °C, and the average of the maximum temperature in the warmest month reaches 35 °C [28]. The Eucalyptus trees planted in this area have shown perfect adaptation and acclimatization to the soil and climatic conditions of the region with no particular maintenance requirements other than fire protection. For each species, five samples harvested from more than five different trees were collected. The identification of specimens was carried out at the Forest Genetic Resources Laboratory by Professors K. Abdelhamid and K. M. Larbi of the National Research Institute for Rural Engineering, Water and Forestry (INRGREF), based on several botanical characteristics described in the literature: young leaves, bark, adult leaves, wood, flowers and fruits, and height as well as the shape of the tree [29,30]. A voucher specimen of each species was deposited at the Laboratory of Management and Valorization of Forest Resources, INRGREF: *E. alba*: ZEA22-4; *E. eugenioides*: ZEE22-3; *E. fasciculosa*: ZEF22-3; *E. robusta*: ZER22-3; and *E. stoatei*: ZES22-1.

Fresh leaves were used for micromorphological, and anatomical analyses, and for the extraction of EOs.

### 2.2. Macromorphological, Micromorphological, and Anatomical Investigation

The leaves were observed by stereomicroscope (LEICA M205 C, Leica Microsystems, Wetzlar, Germany) to assess their general features. Investigations of the main anatomical and micromorphological characteristics were performed by using a Leica DM2000 transmission-light microscope (LM), equipped with ToupCam Digital Camera, CMOS Sensor with a 3.1 MP resolution (ToupTek), and scanning electron microscopy (SEM) Vega3 Tescan LMU SEM (Tescan USA Inc., Cranberry Twp, PA, USA) at an accelerating voltage of 20 kV. Anatomical and micromorphological analyses were carried out both on fresh leaves and on leaves fixed in a 70% ethanol-FineFix solution (Milestone SRL, Sorisole, Bergamo, Italy) for 24 h at 4 °C. Afterwards, the samples were dehydrated in a series of solutions with increasing ethanol content [31]. Healthy and mature leaves were transversally sectioned by hand using a double-edged razor blade. The sections obtained from the fresh leaves were then mounted in water to observe the secretory cavities filled with EO. In addition, sections of thew dehydrated samples were stained with Sudan III to highlight the cuticle thickness, and the distribution of the secretory cavities in the mesophyll. For the better detection of micromorphological features, small leaf specimens (about 0.5 cm^2^) were bleached in a commercial 2.2% sodium hypochlorite solution, overnight. Subsequently, they were washed in water and directly observed or briefly immersed in a diluted acetic acid solution, washed with water, and safranin stained (Merck, Darmstadt, Germany) [32,33]. 

Epidermal surface characteristics and shape, and the distribution of secretory cavities within the mesophyll were also examined by SEM. For the SEM analysis, the fixed and dehydrated samples were critical point dried (CPD, K850 2M Strumenti s.r.l., Rome, Italy), mounted on aluminum stubs, using conductive double-sided adhesive carbon tapes, and finally sputter-coated with a 10 nm layer of gold [34]. 

### 2.3. Extraction of the EOs

The leaves of the five *Eucalyptus* species were separated from the branches and subjected to hydrodistillation for 2 h, in accordance with the method reported by the *European Pharmacopoeia* [35]. The EOs were solubilized in *n*-hexane, filtered over anhydrous sodium sulphate, and stored under N_2_ at +4 °C in the dark until they were tested and analysed.

### 2.4. GC-FID and GC/MS Analyses and Identification of the Essential Oil Components

The composition of the EOs was examined using GC (gas chromatography) and GC-MS (gas chromatography-mass spectrometry) methods. GC analyses were performed using a Perkin-Elmer Sigma 115 gas chromatograph equipped with a flame ionization detector (FID). The analysis was performed using a non-polar HP-5 MS capillary column of fused silica (30 m × 0.25 mm; 0.25 μm film thickness). The operating conditions were as follows: the injector and detector temperatures were 250 and 290 °C, respectively. The analysis was conducted on a scheduled basis: 5 min isothermally at 40 °C; subsequently, the temperature was increased by 2 °C min^−1^ until it reached 270 °C and finally, it was kept in the isotherm for 20 min. The analysis was also performed on a HP Innowax column (50 m × 0.20 nm; 0.25 μm film thickness). In both cases, helium was used as a carrier gas (1.0 mLmin^−1^). GC-MS analysis was performed using Agilent 6850 Ser. II Apparatus equipped with a DB-5 fused silica capillary column (30 m × 0.25 mm; 0.25 μm film thickness) and connected to Agilent Mass Selective Detector (MSD 5973); the ionization voltage was 70 V; the ion multiplier energy was 2000 V. The mass spectra were scanned in the range of 40–500 amu, with five scans per second. The chromatographic conditions were reported as above, and the transfer line temperature was 295 °C. Most of the components were identified by comparing their Kovats indices (Ki) with those in the literature [36,37,38,39] and by a careful analysis of the mass spectra compared to those of pure compounds available in our laboratory or to those present in the NIST 02 and Wiley 257 mass libraries [40]. The Kovats indices were determined in relation to a homologous series of *n*-alkanes (C_10_–C_35_), under the same operating conditions. For some compounds, the identification was confirmed by coinjection with standard samples.

### 2.5. Antioxidant Activity

#### 2.5.1. DPPH Test

The antioxidant activity was determined using the stable 1,1-diphenyl-2-picrylhydrazyl (DPPH) radical method as reported by Brand-Williams and Coworkers [41], with some modifications. The analysis was performed in cuvettes by adding 25 μL of a solution of the EOs in MeOH to 975 μL of a DPPH solution (7.6 × 10^−5^ M), which was prepared daily and kept in the dark to have a final volume of 1 mL in a straight-sided cuvette. Methanol alone was used as a blank, and a cuvette with 1 mL of DPPH solution (60 μM) was used as a control. Absorbance at 515 nm was measured in the spectrophotometer Thermo scientific Multiskan GO (Thermo Fischer Scientific, Vantaa, Finland) after 45 min. The absorbance of DPPH without the antioxidant (control sample) was used for a baseline measurement. The percent inhibition of free radical formation by DPPH (I%) was calculated as follows:I% = ([Ablank − Asample/Ablank]) × 100
where Ablank is the absorbance of the control reaction (containing all reagents except the test compound) and Asample is the absorbance of the test compound read at 515 nm after 45 min. The scavenging activity was expressed as the 50% effective concentration (EC_50_), which is defined as the sample concentration (mg mL^−1^) necessary to inhibit DPPH radical activity by 50% after a 45 min of incubation. Experiments were performed in triplicate and the results are expressed as the mean ± standard deviation.

#### 2.5.2. ABTS Test

The 2,2′-azino-bis 3-ethylbenzothiazoline-6-sulfonic acid (ABTS) test was carried out following the method of Re and coworkers [42]. The ABTS assay and potassium persulfate with a final concentration of 7 and 2.45 mM, respectively, were mixed and left in the dark at room temperature for 16 h before use to produce the radical ABTS (ABTS·+). Trolox (6-hydroxy-2,5,7,8-tetramethylchroman-2-carboxylic acid) was dissolved in methanol at different concentrations and used as a reference standard. The ABTS radical solution was diluted with ethanol to an OD of 0.800 at 734 nm; the absorbance was read at time 0 and 6 min after mixing (Cary Varian, Milano, Italy). The results were expressed as the µM Trolox equivalent antioxidant capacity (TEAC) per gram of samples. All determinations were carried out in triplicate and the results were expressed as the mean ± standard deviation.

### 2.6. Cell Culture

The murine monocyte/macrophage cell line J774A.1 was purchased from American Type Culture Collection (ATCC) and cultured in Dulbecco’s modified Eagle’s medium (DMEM) supplemented with 10% fetal calf serum and cultured at 37 °C in a humidified incubator containing 5% CO_2_. 

### 2.7. Cell Viability Assay 

Cell viability was measured by the 3-[4,5-dimethyltiazol2yl]-2,5 diphenyl tetrazolium bromide (MTT, Sigma-Aldrich, Milano, Italy) assay as previously reported [43]. The J774A.1 cells were seeded on 96-well plates (1 × 10^4^ cells/well) and, the day after, treated with different concentrations (from 3.75 to 120 µg mL^−1^) of Eucalyptus Eos. After 24 h, 25 µL of MTT was added in each well (5 mg mL^−1^ in PBS) and then incubated at 37 °C for 3 h. Afterwards, the dark blue crystals produced were solubilized with DMSO. The optical density of each well was measured at 545 nm using a microplate spectrophotometer reader (Multiskan FC, Thermo Scientific™, Waltham, MA, USA).

### 2.8. Intracellular ROS Measurement

Reactive oxygen species (ROS) were detected by using the fluorescence probe 2′,7′-dichlorofluorescein-diacetate (DCF-DA) as previously reported [44]. First, J774.A1 cells were seeded in 24-well plates (2 × 10^5^ cells/well), then they were treated with the EOs (10 µg mL^−1^ dissolved in dimethyl sulfoxide–DMSO) for 1 h and then stimulated with LPS/IFN-γ (lipopolysaccharide/interferon γ) for 24 h. At the end of the incubation, cells were stained with H2DCF-DA (dichlorodihydrofluoerescein-dicetate) (10 μM) for 30 min at 37 °C. Fluorescence generation was measured by Fluorescent Activated Cell Sorter (FACS) (BriCyte E6, Mindray, China) and analyzed with the FlowJo software (Tree Star, Inc., Elisabeth, NJ, USA). 

### 2.9. Quantification of Nitrite in Cell Culture Supernatants 

The J774A.1 cells were seeded in 24-well plates (2 × 10^5^ cells/well) and treated with the EOs (10 µg mL^−1^ dissolved in DMSO). After 1 h from the treatment, the cells were stimulated with LPS from *Escherichia coli* (O111:B4, Sigma-Aldrich, Milano, Italy) (100 ng mL^−1^) and IFN-γ (20 ng mL^−1^) (Miltenyi Biotec, Bologna, Italy) for 24 h [45]. A standard Griess reaction was performed in duplicate to determine the nitrite concentration. In detail, J774A.1 cell culture supernatants were mixed with Greiss’ reagent (1% sulfanilamide in 5% phosphoric acid and 0.1% N-1-naphthylethylenediamine dihydrochloride in double-distilled water) at a 1:1 ratio. The plate was incubated for 10 min at room temperature and then the absorbance was measured at 550 nm using a microplate photometer reader (Multiskan FC, Thermo Scientific™, Waltham, MA, USA). Absorbance values were interpolated with those of the standard curve generated by a serial dilution of sodium nitrite (1.25–160 µM).

### 2.10. Quantitative Real-Time PCR

The J774A.1 macrophages (5 × 10^5^ cells/well) were treated with the EOs (10 µg mL^−1^) for 1 h before the stimulation with LPS/IFN-γ. After 6 h of incubation, total RNA was extracted using TRI-Reagent (Trizol-Reagent) (Sigma-Aldrich, Milan, Italy), according to the manufacturer’s instructions. Subsequently, cDNA was obtained by reverse-transcription with iScript Reverse Transcription Supermix (Bio-Rad, Segrate, Italy). Quantitative Real-Time PCR (RT-PCR) was performed by using the CFX384 real-time PCR detection system (Bio-Rad). mRNA expression was quantified using specific primers for mouse Gclc, Gclm, and Hmox-1, which are listed below, with the SYBR Green master mix kit (Bio-Rad). Relative gene expression was obtained by normalizing the Ct values of each experimental group against the β-actin transcript level, using the 2-ΔCt formula [46].

The mouse primers were as follows: Gclc: 5′GTTGGGGTTTGTCCTCTCCC-3′; 5′-GGGGTGACGAGGTGGAGTA-3′;Gclm: 5′-AGGAGCTTCGGGACTGTATCC-3′; 5′-GGGACATGGTGCATTCCAAAA-3′; Hmox-1: 5′-GCCGTGTAGATATGGTACAAGGA-3′; 5′-AAGCCGAGAATGCTGAG TTCA-3′.

### 2.11. Statistical Analysis

Statistical analysis was performed using the GraphPad Prism software version 9 (San Diego, CA, USA). For the comparison of two groups, a t-test was used, while for the comparison of multiple groups, the ANOVA test was used. The data were shown as mean ± SEM. A *p* value of < 0.05 was considered statistically significant and was labeled with *; *p* values < 0.01, 0.001 or 0.0001 were labeled with **, *** or ****, respectively.

## 3. Results

### 3.1. Macromorphological, Micromorphological, and Anatomical Investigation

The main macromorphological and micromorphological features of the leaves of the five species are reported in Table 1.

All five species of *Eucalyptus* examined had adult leaves with different shapes and characteristics (Figure 1A–E). In *E. alba*, they were thin, and generally lanceolate with a pointed apex (Figure 1A). In *E. eugenioides*, the leaves were lanceolate-falcate, and had a pointed apex (Figure 1B); *E. fasciculosa* showed broadly lanceolate to ovate leaves, with the base being generally oblique (Figure 1C). *E. robusta* leaves were broadly lanceolate and slightly falcate, with a pointed apex (Figure 1D); *E. stoatei* had coriaceous-leathery leaves, which were elliptical to oblong, with an apiculate to mucronate apex (Figure 1E). A prominent intramarginal vein near the margin, and running more or less parallel to it, was present in the leaves of all species (Figure 1F, referring to *E. robusta*). 

All species were amphistomatic and showed an isobilateral mesophyll structure, with a multilayered palisade parenchyma within which many secretory cavities were found (Figure 2A–F and Figure 3A–E), that were mainly spherical in shape (Figure 2B and Figure 3A–G).

The secretory cavities are the site of production and accumulation of the essential oil, as shown by the epidermal peeling after being bleached with sodium hypochlorite (Figure 3F), where it appeared bright yellow, due to the richness in EO. In addition, also in the same samples, the overlying cells located over the secretory cavities were sometimes visible (Figure 3F, arrows). In the hand-made transversal section of the fresh leaf, mounted in water, the brownish yellow drops of essential oil could be detected within a secretory cavity (Figure 3G).

The leaf epidermal surface of all the studied species showed a thick cuticle with abundant wax and cutin depositions, orange-red-stained by Sudan III (Figure 3A–E,H). Druses or prismatic crystals of calcium oxalate were abundant in the mesophyll of all species, mainly located near the veins and the epidermis (Figure 3A–E,H, arrow).

The pink-red staining with safranin allowed us to highlight the overlying cells (Figure 4A,C,E,G,I), and the SEM analysis showed the presence of more or less prominent papillae (Figure 4B,D,F,H,J) which were sometimes not clearly visible, probably due to the abundance of wax and cutin depositions in the cuticle (Figure 4D,H). Overlying cells associated with secretory cavities differed from ordinary cells in shape, size and/or color. The difficulty in identifying them was due to the fact that the secretory cavities were located deep in the mesophyll (Figure 2), and also due to the thick cuticle layer (Figure 3A–E). Only after bleaching, and safranin staining, could the overlying cells become visible in all leaves, except for those of *E. eugenioides*, in which only a depression zone in correspondence with the secretory cavity was observed (Figure 4C). *E. robusta* was the only species displaying up to four overlying cells (Figure 4G), while only two cells were observed in the other species (Figure 4A,E,I). Both actinocytic (Figure 4K) and anomocytic stomata (Figure 4L) were found. In some cases, stomata were deeply sunken below the leaf surface, as shown in the leaves of *E. stoatei* (Figure 3H, arrow).

### 3.2. Chemical Composition of Essential Oils

The hydrodistillation gave yellow pale EOs, in 1.06, 1.66, 0.99, 2.01, and 1.92% yields on a dry weight basis, respectively, for *Eucalyptus alba*, *E. eugenioides*, *E. fasciculosa*, *E. robusta*, and *E. stoatei*. Table 2 reports the chemical composition of the five Eucalyptus EOs. The components are listed according to the elution order in a HP-5 column.

Altogether, 31 components were identified, 11 in the *E. alba* EO, accounting for 96.3% of the total, 2 in the EO of *E. eugenioides* (97.5%), 16 in the EO of *E. fasciculosa* (93.8%), 8 in the EO of *E. robusta* (94.6%) and 10 in the EO of *E. stoatei* (94.4%). In all samples, except in the *E. alba* EO, oxygenated monoterpenes predominated, at percentages between 65.1 (*E. robusta* EO) and 95.9% (*E. eugenioides* EO). In the EO of *E. alba*, monoterpene hydrocarbons constituted the main class, at 58.4%. Appreciable amounts of sesquiterpenes, above all sesquiterpene hydrocarbons, were found in the EOs of *E. fasciculosa*, *E. robusta*, and *E. stoatei* only. Except for the EO of *E. alba*, in which α-pinene was the main constituent (54.1%), eucalyptol was predominant in the EOs, at percentages between 64.4 (EO of *E. robusta*) and 95.9% (EO of *E. eugenioides*). The EO of *E. alba* in addition to the high amount of α-pinene, showed eucalyptol at 25.6%, and other components were present in appreciable amounts such as *cis*-β-guaiene (4.1%), camphene (3.9%) and borneol (3.4%). Other EO component contents were less than 3%. Only two components were detected in the EO of *E eugenioides*: eucalyptol (95.9%) and 1,3,8-*p*-menthatriene (1.6%). The EO with the greatest number of components was instead that obtained from *E. fasciculosa*: in addition to eucalyptol (72.8%), the main components were viridiflorene (6.1%), aromadendrene (3.7%), *allo*-aromadendrene (2.8%) and dehydro-aromadendrene (2.7%); other components were less than 2%. In the EO from *E. robusta*, in addition to eucalyptol (64.4%), cadinenes accounted for 19.6% of the yield, with a predominance of δ-cadinene (15.7%). Apart from α-muurolene (6.4%), the others components were present in amounts of less than 3%. In the *E. stoatei* EO, in addition to eucalyptol (71.1%), the main components were α-pinene (13.6%) and aromadendrene (4.7%); other components were present in amounts of less than 3%.

### 3.3. Antioxidant Activity by the DPPH Assay

The antioxidant activity determined by the DPPH assay is reported in Table 3. 

The results show the strong antioxidant activity of the EOs. The EO from *E. alba* was the most active, with an IC_50_ of 1.70 ± 0.79 mg mL^−1^, followed by the EO of *E. robusta*, which had an IC_50_ of 2.05 ± 0.97 mg mL^−1^. The EOs from *E. eugenioides* and *E. fasciculosa* were also active, albeit to a lesser extent, with IC_50_ values of 4.93 ± 1.74 and 7.82 ± 2.69 mg mL^−1^, respectively. Finally, the least active was the EO of *E. stoatei* (IC_50_: 45.62 ± 1.25 mg mL^−1^).

### 3.4. Antioxidant Activity by the ABTS Assay

The antioxidant activity determined by the ABTS assay is reported in Table 4. 

The analyses showed that, overall, the five EOs exhibited excellent antioxidant activity, with Trolox value equivalents (TE) ranging between 9.3 µM gr^−1^ (*E. stoatei* EO) and 25.8 µM gr^−1^ TE (*E. eugenioides* EO). The latter was more active; the EOs of *E. alba, E. fasciculosa* and *E. robusta* exhibited similar activity. Additionally, in this test, the EO of *E. stoatei* was the least active.

### 3.5. Effect of the EOs on Cell Vitality

To better characterize the antioxidant effects of the EOs, the cell line J774A.1 was used. First, the evaluation of the possible cytotoxic effect exerted by the EOs was performed by a MTT assay. The J774A.1 murine macrophages were treated with increasing concentrations (from 3.75 to 120 µg mL^−1^) of the five EOs (EA = *E. alba,* EG *= E. eugenioides,* EF *= E. fasciculosa,* ER *= E. robusta* and ES *= E. stoatei*). for 24 h. As shown in Figure 5, all the EOs did not significantly affect cell viability, except for the ER EO at 120 µg mL^−1^. Thus, the concentration of 10 µg mL^−1^ was selected for the following studies.

### 3.6. Activity on ROS Production and Modulation of the Expression of Antioxidant Enzymes

Once activated by external threats, macrophages differentiate into the M1 phenotype, which produce a series of cytokines, ROS and other inflammatory mediators in order to neutralize the inciting cause [44]. To gain further insight into the antioxidant activity of the EOs, we evaluated the generation of ROS using the DCF-DA fluoroprobe (Figure 6A). In particular, J774A.1 cells were pretreated with the EOs (10 μg mL^−1^ dissolved in DMSO) for 1 h and then stimulated with LPS and IFN-γ to induce the proinflammatory phenotype M1. As expected, the stimulation enhanced the production of ROS while the pretreatment markedly suppressed it, demonstrating the antioxidant effect of the EOs (Figure 6B). To corroborate these findings, the gene expression of phase II enzymes, such as glutamate-cysteine ligase (GCL) and heme oxygenase-1 (Hmox-1), that play a key role in redox homeostasis and the suppression of oxidative stress, were evaluated. In line with the previous results, the pretreatment with the EOs resulted in an increase in Gclc, Glcm and Hmox-1 expression levels (Figure 6C). 

### 3.7. Activity of the EOs on NOx Production in M1 Murine Macrophages

To test whether the EOs exerted an anti-inflammatory effect, we evaluated their ability to inhibit the production of nitric oxide, one of the main inflammatory mediators [45]. To assess it, we pretreated J774A.1 cells with the five EOs at a concentration of 10 µg mL^−1^ and subsequently, after 1 h, the cells were stimulated with LPS and IFN-γ. After 24 h of incubation, a Griess assay was performed; the pretreatment with all EOs, except ES, significantly reduced nitric oxide production (Figure 7).

## 4. Discussion

Some taxa of *Eucalyptus* are indeed difficult to distinguish, and Eucalyptus leaf morphology can provide a range of diagnostic features. For this reason, it is important to perform macromorphological, micromorphological and anatomical investigations of the leaves to confirm the identity of the samples. 

Our observations are in agreement with previous data reporting that Eucalyptus leaves are generally isobilateral, and amphistomatic, with a multilayered palisade, in which many oxalate crystals are spread [33]. In addition, similarly to what has been observed in the leaves of *E. cinerea* F. Muell. ex Benth. [47], we found anomocytic stomata that are typical in the *Eucalyptus* genus [33,47,48,49], but also actinocytic stomata that are instead more common in other Myrtaceae genera [50]. 

The xerophytic leaves of all examined species showed a thick to very thick and waxy cuticle and sunken stomata as adaptations to very hot and dry conditions. Differently to other *Eucalyptus* species [18,19,48], the secretory cavities located within the mesophyll could only rarely be found in the subepidermal region of the two faces of the leaf. All these features made it difficult to identify the modified epidermal cells covering the secretory cavities. These overlying cells are made of two to four cells, generally differing in shape, dimension and sometimes color from the epidermal ones. Similar observations were also referred to by Santos and coworkers [48], who described two overlying cells in six *Eucalyptus* species and up to four overlying cells only in *E. pyrocarpa* L.A.S. Johnson & Blaxell. However, in our study, a depression zone over the secretory cavities was distinguishable, but no overlying cells were distinguishable in the *E. eugenioides* leaf epidermis.

The Eucalyptus leaves produced pale EOs with a yield between 0.99 (*E. fasciculosa*) and 2.01% (*E. robusta*), in accordance with the heterogeneous oil yield reported in the literature for Eucalyptus plants.

Some studies have reported the composition of the EO of *E. alba*, even though only one reported that from plants grown in Tunisia. Elaissi and coworkers [51] reported an EO rich in eucalyptol (44.1%) and appreciable amounts of sesquiterpenes, such as spathulenol (5.8%), globulol (4.9%), and *trans*-caryophyllene; this composition agrees in part with data recorded in the present study. Our composition generally is consistent with the chemical profiles of the EO of *E. alba* grown in other countries: Bangladesh [52], Congo [53], Senegal [54,55], and Burkina Faso [56].

The composition of the EO of *E. eugenioides* found by us differs completely from that reported by the only study present in the literature. In fact, Bignell and coworkers [57] reported the presence of a much greater number of components, over 50, among which α-pinene (22%) and β-eudesmol (10%) were the main constituents. These compounds were absent in our sample, in which only the presence of eucalyptol and 1,3,8-*p*-menthatriene was detected. 

Some papers reported the composition of the EO of *E. fasciculosa* [58,59,60]. These studies highlighted a much richer composition than the one examined in this work. The concordance is found in the presence of eucalyptol as the main component with percentages higher than 50% and in the significant presence of spathulenol, sometimes in higher quantities (around 10%) than those found in our sample. Differently from the literature, α-pinene, limonene, *p*-cymene, α-terpineol, viridiflorol, caryophyllene, and globulol, reported as being among the main components of the EO, were scarce or absent in our sample. We instead found that this EO was rich in viridiflorene, which was absent in the cited papers. 

The available literature reports a series of studies on the composition of the EO of *E. robusta* [53,59,60,61,62,63,64,65,66,67,68,69,70]. These studies described EO compositions that were different to those of our sample. Eucalyptol remains the main component in all cases, even if it found in smaller quantities that do not exceed 30%. Furthermore, in some scientific papers, α-pinene is the main component, and appreciable amounts of *p*-cymene, limonene, spathulenol, globulol and α-terpineol are reported. All these compounds were instead absent in our sample.

The available literature includes few studies concerning the composition of the EO of *E. stoatei* [71,72]. These studies referred to compositions mainly rich in α-pinene, eucalyptol and aromadendrene. Although the concentrations are different, these data agree with the composition of our sample.

Considering their composition, four of the EOs can be attributed to the eucalyptol chemotype; in fact, for *Eucalyptus* species, the presence of chemotypes has been proposed [73]. On the other hand, the chemical composition of the EOs depends on different intrinsic and extrinsic factors; firstly, the species is one factor, and secondly, the age of the plant and the chemical and physical conditions of the growth environment and of harvesting (season, location, climate, soil, and developmental stage) are another factor. 

The DPPH assay has been employed in evaluating the antioxidant activity of some Eucalyptus EOs [27,74,75]. However, there are no literature reports in which this test was performed on the EOs of *E. alba*, *E. eugenioides*, *E. fasciculosa* and *E. stoatei*. Only the paper of Cahaya and coworkers [76] reported the antioxidant activity of an EO of *E. robusta*, with a weaker activity in comparison to that of our sample.

In any case, the composition reflects the antioxidant and anti-inflammatory activities of the EOs studied. In fact, in the *E. alba* EO, the main components are α-pinene and eucalyptol, which are probably responsible for the antioxidant activity of the whole EO. In fact, the antioxidant activity of α-pinene has been evaluated through various assays, including the DPPH test [77,78,79]. Eucalyptol has also been studied for its antioxidant action [1,80,81,82,83]. However, these studies also showed that these monoterpenes alone do not have sufficient antioxidant power. Therefore, it is possible that the whole activity is due to a synergistic action between these and other components of the EO, such as camphene [84], borneol [85], *cis*-β-guaiene [86], and spathulenol [87].

Our EO of *E. eugenioides* almost exclusively constituted eucalyptol, to which the antioxidant activity can be attributed. 

Some components detected in the EO of *E. fasciculosa* have been reported for their antioxidant activity: α-phellandrene [88], aromadendrenes [89], viridiflorene [90], and spathulenol [91].

Apart from eucalyptol, some other components of the EO of *E. robusta* have been evaluated for their antioxidant activity: β -copaene [92], α-muurolene and cadinenes [93].

Our EO from *E. stoatei* was characterized by some components known for their antioxidant properties, such as eucalyptol, α-pinene, eucalyptol, aromadendrene, and viridiflorene [90].

Moreover, our results confirmed the antioxidant activity of the EOs, even compared to other Eucalyptus EOs. It has also been reported that a particular technique based on a leaf grinding process followed by sieving fractionation increased the antioxidant phytochemicals in the EO of some *Eucalyptus* species [94]. Overall, our EOs, exhibiting vigorous antioxidant activity, can protect against oxidative diseases and can be used as natural antioxidants in the food and confectionery industries [95]. 

Although the available literature reports that the antioxidant activity of Eucalyptus EOs is linked to their monoterpenes [96], from our data, it was not possible to identify the specific active components. For example, the EOs with the highest and the weakest antioxidant activity (those of *E. eugenioides* and *E. stoatei*, respectively)both contain high percentages of eucalyptol (95.9% and 71.1%, respectively). Probably, the absence of α-pinene in the EO of *E. stoatei* caused the loss of antioxidant activity. However, the EO of *E. alba*, with good antioxidant activity, contained 54.1% of α-pinene, whereas the EOs from *E. robusta* and *E. fasciculata*, with similar antioxidant activity, did not contain this compound. Furthermore, the antagonistic activity between different components of Eucalyptus EOs was observed by Ciesla and coworkers [97], who identified solid antagonistic activity between the binary mixture of eucalyptol—*p*-cymene. When present as major components, these resulted in weak antioxidative activity [98]. 

On the whole, our results agree with those of the recent literature on the chemical composition and the biological activities of Eucalyptus EOs [99], even if chemical and biological investigation are needed, due to the complex chemodiversity in the genus and also in some *Eucalyptus* species [100].

In the available literature, few studies have reported the effects of Eucalyptus EOs on the cell viability and antioxidant activity of cells. These studies mainly concerned *E. citriodora* Hook., *E. bridgesiana* F. Muell. ex F.T. Baker, *E. globulus* and *E. teritecornis* Sm. [101], but no studies are available on the species considered in the present work. However, analyzing the available literature, some points deserve to be underlined. A study, conducted on *E. globulus*, using the J774A.1 murine macrophage cells, showed that a pretreatment with *E. globulus* extracts significantly inhibited inducible nitric oxide synthase (iNOS) mRNA expression [102]. In another work, the activity was evaluated by the 5-LOX (5-Lipoxigeanse) inhibition assay, and by the proinflammatory cytokine secretion assay which; however, this study did not show particular activity in the EO from *E. citriodora* [103]. The EOs from *E. globulus, E. citriodora* and *E. teritecornis* have been reported for certain analgesic and anti-inflammatory activities [23,103]. The cell viability evaluated by a MTT assay, the NO production assay and Western blotting revealed the activation of NF-kB (Nuclear Factor kappaB) by the EO from *E. citriodora*, which showed its excellent inhibition activity in the production of proinflammatory molecules including iNOS [104]. An assay of the cyclooxygenase activity of PGHS (prostaglandin H synthase) and another assay of lipoxygenase-L-1 activity resulted in good inhibitory activity for the EO of *E citriodora* in the last enzyme [105]. Moreover, the excellent activity of the EO of *E. globulus* in the inhibition of albumin denaturation was also reported [106].

In line with these findings, our study demonstrated the promising antioxidant and anti-inflammatory activities of Eucalyptus EOs. In fact, our results showed that the EOs were able to thwart the proinflammatory function of M1 macrophages, as observed by the reduction in NO production, an important mediator and regulator of the inflammatory response. Likewise, all the EOs tested showed great antioxidant activity, as demonstrated by using the 2′,7′-dichlorodihydrofluorescein diacetate (H2DCFDA) probe. In addition, these EOs significantly increased the catalytic (GCLC) and the modulatory (GCLM) subunits of glutamate-cysteine ligase (GCL), the key enzyme involved in glutathione (GSH) synthesis. Moreover, the EOs increased the expression level of the antioxidant enzyme Hmox-1, highlighting their ability to reduce ROS production.

## 5. Conclusions

The present study was focused on the EOs of five *Eucalyptus* species, namely *Eucalyptus alba*, *E. eugenioides*, *E. fasciculosa*, *E. robusta*, and *E. stoatei*. Four of these EOs can be attributed to the eucalyptol chemotype, while that of *E. alba* showed α-pinene as the main component. The antioxidant and anti-inflammatory properties of these EOs suggest that they may be a promising therapeutic strategy for inflammation-based diseases. The growing awareness of the presence of bioactive molecules, especially in their volatile fractions, makes the plants of this genus a solid starting point for obtaining useful products in the nutraceutical and pharmaceutical industries.

## Figures and Tables

**Figure 1 antioxidants-12-00867-f001:**
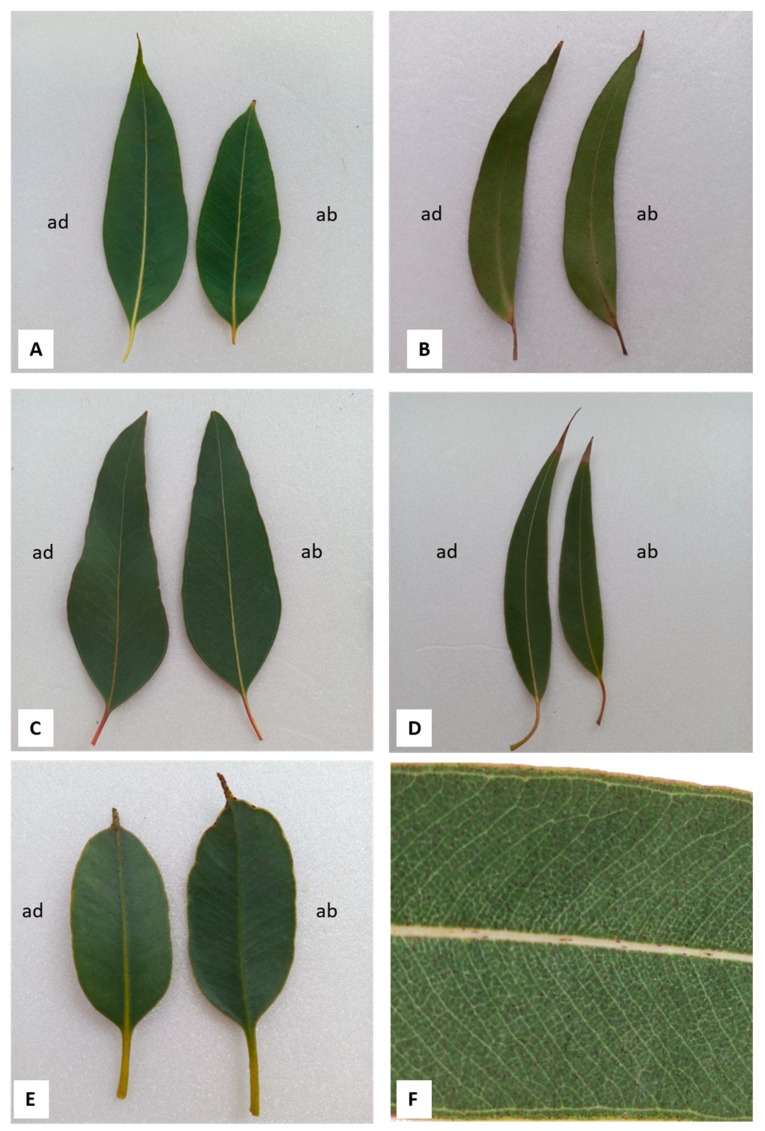
Morphology of leaves of the five *Eucalyptus* species: *E. alba* (**A**), *E. eugenioides* (**B**), *E. fasciculosa* (**C**)*, E. robusta* (**D**), and *E. stoatei* (**E**); example of intramarginal veins running parallel with the leaf margin in *E. robusta* (**F**). (ad, adaxial side; ab, abaxial side).

**Figure 2 antioxidants-12-00867-f002:**
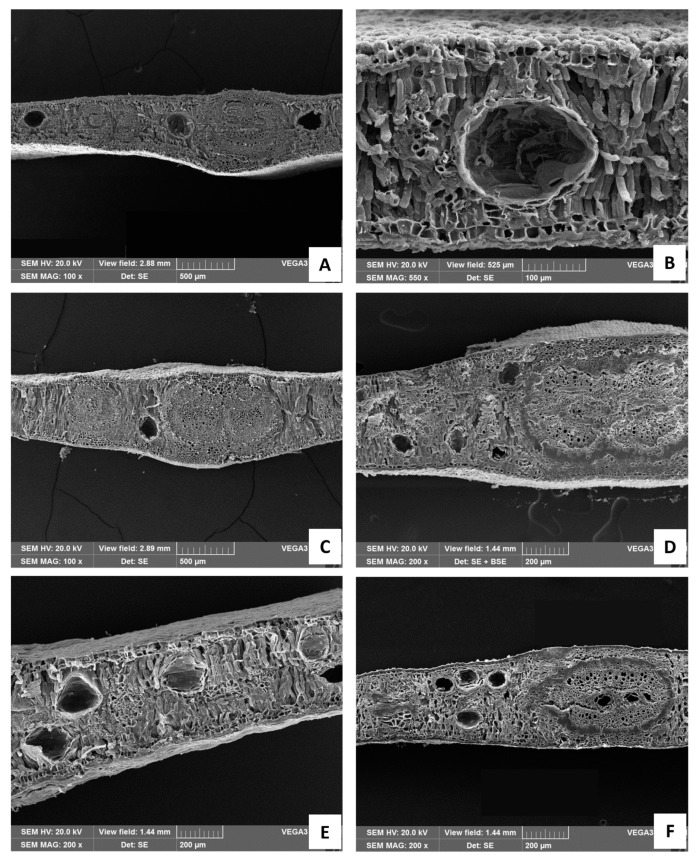
SEM micrographs of the transverse leaf sections of the five *Eucalyptus* species (**A**–**F**). The secretory cavities within the mesophyll are clearly visible in *E. alba* (**A**), and one secretory cavity at higher magnification is visible in the leaf of *E. alba* (**B**); *E. eugenioides* (**C**), *E. fasciculosa* (**D**), *E. robusta* (**E**), and *E. stoatei* (**F**).

**Figure 3 antioxidants-12-00867-f003:**
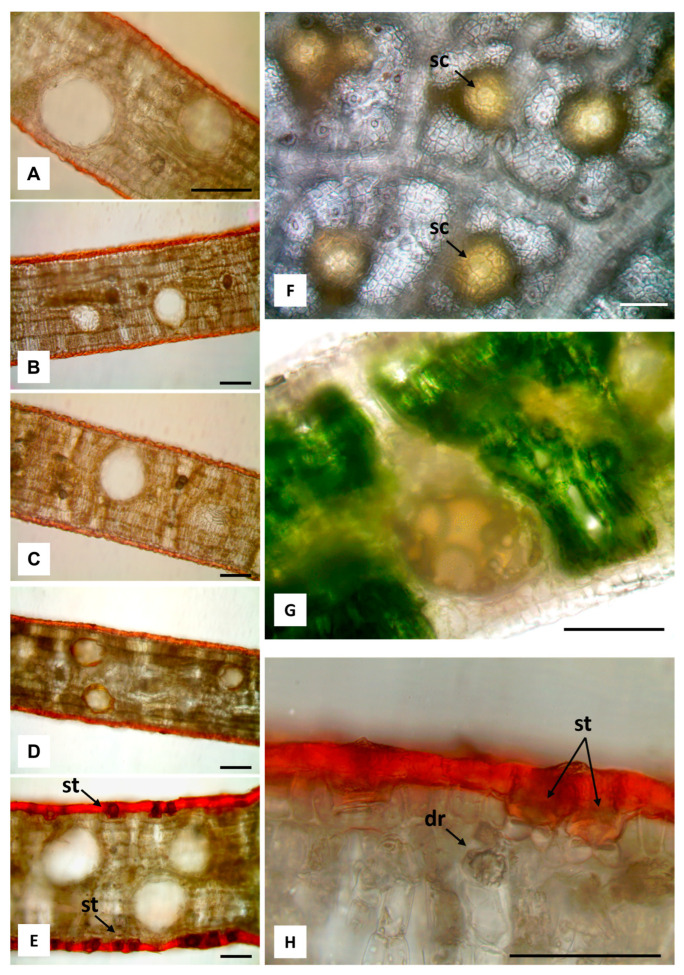
Light microscopy images. (**A**–**E**) Hand-made transverse leaf sections stained with Sudan III highlight the thick layers of cuticle and the secretory cavities spread in the mesophyll: *E. alba* (**A**), *E. eugenioides* (**B**), *E. fasciculosa* (**C**), *E. robusta* (**D**), *E. stoatei* (**E**). (**F**,**G**) Leaves of *E. alba*: abaxial surface after bleaching with sodium hypochlorite, showing bright yellow secretory cavities (sc), and their overlying cells (arrows) (**F**); transverse section showing a secretory cavity containing many drops of brownish yellow EO (**G**). (**H**) Transverse section of *E. stoatei* leaf stained with Sudan III, showing a thick cuticle, and stomata (st); a druse (dr) under the epidermis is also visible. 100 µm bars.

**Figure 4 antioxidants-12-00867-f004:**
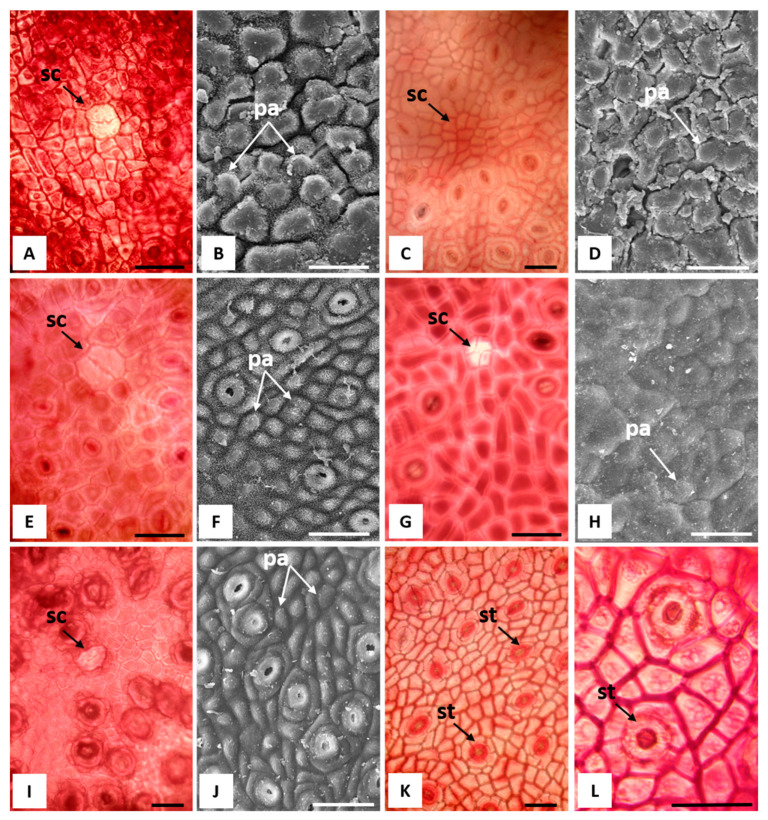
LM and SEM images: *E. alba* (**A**,**B**), *E. eugenioides* (**C**,**D**,**K**), *E. fasciculosa* (**E**,**F**), *E. robusta* (**G**,**H**,**L**), and *E. stoatei* (**I**,**J**). LM: (**A**,**C**,**E**,**G**,**I**)—leaf epidermis stained with safranin and showing overlying cells covering the secretory cavities. SEM: (**B**,**D**,**F**,**H**,**J**)—leaf surface with more (**B**,**F**,**J**) or less (**D**,**H**) evident papillae. LM: (**K**,**L**)—two types of stomata, actinocytic and anomocytic, are shown; actinocytic stomata in the epidermis of *E. eugenioides* (**K**, arrows), and anomocitic stomata in the epidermis of *E. robusta* (**L**, arrow). Overlying cells (sc) in correspondence with the secretory cavities, papillae (pa), and stomata (st). 50 µm Bars.

**Figure 5 antioxidants-12-00867-f005:**
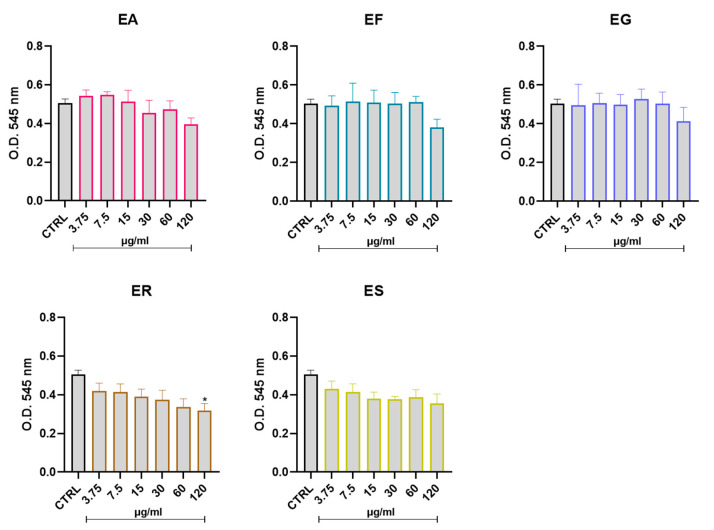
Effect of the EOs on cell vitality. The J774A.1 macrophages were treated with increasing concentrations of Eos, and cell viability was examined after 24 h by the MTT assay. Values are expressed as mean ± SEM from three independent experiments. * *p* < 0.05 indicates the significant effect of the EOs compared to control cells (EA = *E. alba*, EG = *E. eugenioides,* EF = *E. fasciculosa*, ER = *E. robusta* and ES = *E. stoatei*).

**Figure 6 antioxidants-12-00867-f006:**
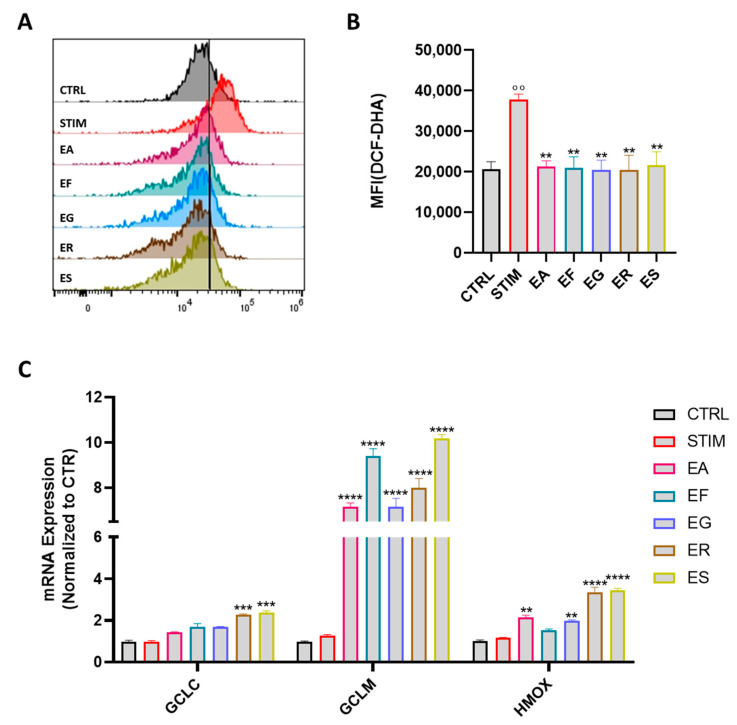
Effect of the Eucalyptus EOs on oxidative stress. J774A.1 macrophages pretreated with the EOs before being stimulated with LPS and INFγ. Representative histograms and relative quantification of DCF-DA fluorescence in J774A.1 cells (**A**,**B**). Relative mRNA levels of gclc, gclm and hmox-1 in J774A.1 macrophages determined by RT-PCR analysis after 6 h (**C**). Values are expressed as mean ± SEM from three independent experiments. °° *p* < 0.01 indicates significant effect of LPS/INF-γ-stimulated cells (STIM) compared to unstimulated cells (CTRL); ** *p* < 0.01, *** *p* < 0.001, **** *p* < 0.0001 indicate significant effect of Eucalyptus extracts compared to stimulated cells. EA = *E. alba*, EG = *E. eugenioides*, EF = *E. fasciculosa*, ER = *E. robusta* and ES = *E. stoatei*.

**Figure 7 antioxidants-12-00867-f007:**
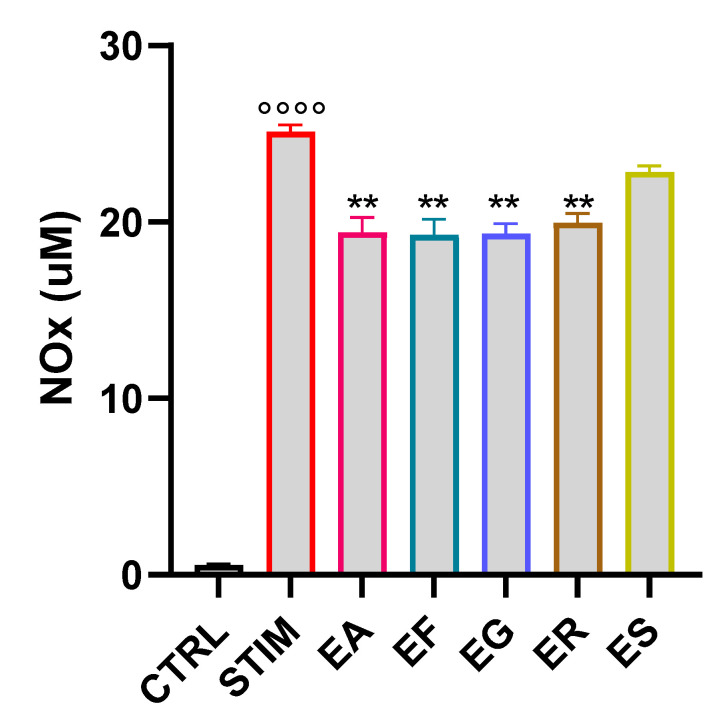
Eucalyptus Eos’ inhibition of NOx production. The J774A.1 macrophages were pretreated with the EOs (10 µg mL^−1^ dissolved in DMSO) for 1 h before being stimulated with LPS (100 ng/mL) and INFγ (20 ng mL^−1^). NO levels were measured in the cell culture medium after 24 h by the Greiss reaction. Values are expressed as mean ± SEM from three independent experiments. °°°° *p* < 0.0001 indicates the significant effect of LPS/INF-γ-stimulated cells compared to unstimulated cells (CTRL); ** *p* < 0.01 indicates a significant effect of the EOs compared to stimulated cells. EA = *E. alba*, EG = *E. eugenioides*, EF = *E. fasciculosa*, ER = *E. robusta* and ES = *E. stoatei*.

**Table 1 antioxidants-12-00867-t001:** Main macromorphological and micromorphological characteristics of the leaves of the *Eucalyptus* species studied.

Leaf Features	*E. alba*	*E. eugenioides*	*E. fascicolosa*	*E. robusta*	*E. stoatei*
Leaf size (Length × width in cm) n = 10	9.7–15.4 × 2.3–3.7	7.5–10.4 × 1.6–2.3	9.9–11.7 × 3.0–4.8	12.3–16.3 × 1.9–2.9	5.7–7.3 × 2.5–3.1
Leaf shape	Lanceolate	Lanceolate/ falcate	Broadly lanceolate/ovate	Broadly lanceolate/slightly falcate	Elliptical/oblong
Texture	Papery	Papery/Coriaceous	Coriaceous	Coriaceous	Coriaceous/Leathery
Stomata	Actinocytic	Actinocytic/ anomocytic	Anomocytic	Anomocytic	Anomocytic
Wax and cutin depositions in the cuticle	++	+++	++	+++	+++
Papillae	Present	Scarcely visible	Present	Scarcely visible	Present

++ = abundant; +++ = very abundant.

**Table 2 antioxidants-12-00867-t002:** Chemical composition of the EOs.

	*E. alba*	*E. eugenioides*	*E. fasciculosa*	*E. robusta*	*E. stoatei*	Ki ^a^	Ki ^b^	Identification ^c^
α-Pinene	54.1	-	0.1	-	13.6	857	1036	1,2,3
Camphene	3.9	-	-	-	-	869	1075	1,2,3
Thuja-2,4(10)-diene	-	-	-	-	0.2	875	1115	1,2
β-Pinene	0.4	-	-	-	0.1	894	1136	1,2,3
α-Phellandrene	-	-	1.5	-	-	922	1177	1,2,3
α-Terpinene	-	-	0.6	-	-	933	1170	1,2,3
Eucalyptol (1,8-cineole)	25.6	95.9	72.8	64.4	71.1	943	1210	1,2,3
*cis*-Sabinene hydrate	-	-	-	0.7	-	955	1070	1,2
1,3,8-*p*-Menthatriene	-	1.6	-	-	-	958	-	1,2
*exo*-Fenchol	0.3	-	-	-	-	1019	1591	1,2
Borneol	3.4	-	-	-	-	1067	1615	1,2,3
β-Panasinsene	-	-	0.1	-	0.1	1275	-	1,2
α-Gurjunene	-	-	0.3	-	-	1288	1535	1,2
(Z)-Caryophyllene	-	-	1.0	-	-	1296	1617	1,2
β-Cedrene	-	-	0.1	0.1	-	1298	1625	1,2
β-Copaene	-	-	0.1	2.7	-	1304	1628	1,2
Aromadendrene	0.8	-	3.7	-	4.7	1308	1631	1,2
β-Longipinene	-	-	-	0.7	-	1323	-	1,2
*allo*-Aromadendrene	0.8	-	2.8	-	0.9	1330	1660	1,2
Dehydro-Aromadendrene	-	-	2.7	-	-	1331	1642	1,2
γ-Gurjunene	1.7	-	-	-	-	1343	1687	1,2
δ-Selinene	-	-	-	-	0.2	1350	1707	1,2
γ-Amorphene	-	-	0.6	-	-	1358	1695	1,2
*cis*-β-Guaiene	4.1	-	-	-	-	1361	1748	1,2
*trans*-β-Guaiene	-	-	-	-	0.6	1366	1723	1,2
Viridiflorene	-	-	6.1	-	2.9	1367	1713	1,2
α-Muurolene	-	-	-	6.4	-	1373	1744	1,2
γ-Cadinene	-	-	-	3.9	-	1385	1752	1,2
δ-Cadinene	-	-	-	15.7	-	1396	1755	1,2
Germacrene B	-	-	0.2	-	-	1433	1795	1,2
Spathulenol	1.2	-	1.1	-	-	1452	2127	1,2
Total	96.3	97.5	93.8	94.6	94.4			
Monoterpene hydrocarbons	58.4	1.6	2.2	-	13.9			
Oxygenated monoterpenes	29.3	95.9	72.8	65.1	71.1			
Total monoterpenes	87.7	97.5	75.0	65.1	85.0			
Sesquiterpene hydrocarbons	7.4	-	17.7	29.5	9.4			
Oxygenated sesquiterpenes	1.2	-	1.1	-	-			
Total sesquiterpenes	8.6	-	18.8	29.5	9.4			
Ratio of monoterpenes/sesquiterpenes	10.2	-	4.0	2.2	9.0			

^a,b^ The Kovats retention indices are relative to a series of *n*-alkanes (C10–C35) in the apolar HP-5 MS and the polar HP Innowax capillary columns, respectively. **^c^** Identification method: 1 = comparison of the Kovats retention indices with published data, 2 = comparison of mass spectra with those listed in the NIST 02 and Wiley 275 libraries and with published data, and 3 = coinjection with authentic compounds; t = trace (<0.1%). - = absent.

**Table 3 antioxidants-12-00867-t003:** Antioxidant activity of the EOs according to the DPPH assay. Data are expressed as the mean ± SD of three experiments.

Essential Oil	IC_50_ (mg mL^−1^)
*E. alba*	1.70 ± 0.79
*E. eugenioides*	4.93 ± 1.74
*E. fasciculosa*	7.82 ± 2.69
*E. robusta*	2.05 ± 0.97
*E. stoatei*	45.62 ± 1.25

**Table 4 antioxidants-12-00867-t004:** Antioxidant activity according to the ABTS assay. Data are expressed as the mean ± SD of three experiments.

Essential Oil	TEAC (µM gr^−1^)
*E. alba*	21.5 ± 2.28
*E. eugenioides*	25.8 ± 1.39
*E. fasciculosa*	21.2 ± 1.45
*E. robusta*	22.5 ± 0.39
*E. stoatei*	9.3 ± 1.77

## Data Availability

The data presented in this study are available upon request from the corresponding author.

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
