# Peer review of "Essential Oil Composition, Antioxidant Activity and Leaf Micromorphology of Five Tunisian Eucalyptus Species"

_antioxidants, 2023, doi:10.3390/antiox12040867_

Round 1

Reviewer 1 Report

The manuscript presents results of extensive comparative research on leaf anatomy, composition, and antioxidant activity of essential oil of five Eucalyptus species. The results have theoretical and practical importance due to the use of eucalyptus oil in medicine and cosmetics.

Comments:

Abstract:  there are no Latin names of the analyzed species in this section.

Material and methods: 

In this section and in other parts of the manuscript, the authors claim that they studied the micromorphology of leaves, which is a misconception, as the investigations conducted based on transversal sections of leaves present their anatomy. Micromorphological analyses are mainly carried out using SEM and are usually focused on details of the epidermis structure, e.g. the cuticle surface sculpture, trichomes, and location of stomata.

Results

L. 319. The authors claim that there are “prominent papillae” in the epidermis of some Eucalyptus species, but these structures are not visible in the photographs and have not been marked. It seems improbable that the leathery Eucalyptus leaves, which have a thick cuticle layer, bear papillae.

In the description of figure 3, the “E”, “H”, and explanation of the “st” abbreviation are missing. In turn, stomata are very clearly visible in photograph 3H, which is worth pointing out.

In table 2 showing the components of essential oil, the groups of monoterpenes and sesquiterpenes given at the end of the table should be indicated/connected.

The text requires linguistic proofreading.

Reviewer 2 Report

Article:   Essential oil composition, antioxidant activity and leaf micromorphology of five Tunisian Eucalyptus species

The idea of the manuscript is good, new and interesting, written in a great way, with clear and distinctive data Abstracts: good. Keywords: add " species " Introduction: The introduction focused on the properties and benefits of camphor trees only, and the authors neglected the main paragraph that explains the most important varieties of camphor, and the differences between them. So the introduction needs more support. Materials and Methods: "Leafy branches of five species of eucalyptus, namely Eucalyptus alba Reinw. ex Blume, 105 E. eugenioides Sieber ex Spreng., E. fasciculosa F. Muell., E. robusta Sm., and E. stoatei C. 106 Gardner, were collected", this paragraph needs a more comprehensive description, for example, the age of the trees, the places of their cultivation, the planting distances, and the methods of caring for them. Results: The authors presented the results in a distinct and clear manner Discussion: The authors mentioned some taxa of Eucalyptus are indeed difficult to distinguish, and eucalyptus leaf morphology can provide a range of diagnostic features. For this reason, it is important to perform macro and micro-morphological analysis of the leaves to confirm the identity of the samples, the most important question here is on what basis the authors mentioned the scientific names of the five taxa at the beginning of the study.

Conclusions:  mention should be made of the species used in the study.

Reviewer 3 Report

Reviewer comments

Journal: Antioxidants (ISSN 2076-3921)

Manuscript IDantioxidants-2279242

Manuscript Title: “Essential oil composition, antioxidant activity and leaf micromorphology of five Tunisian Eucalyptus species”

In the current study, the authors wrote an article about EOs constituent, antioxidant activity and leaf micromorphology of five Tunisian Eucalyptus species.

The topic is nice and fits well with the scope of Antioxidants-MDPI, and the results are of interest to the scientific community. The methods are clear and sound. The figures are prepared in high quality, while the results and discussions have to merge. However, the text needs a major revision before publication in Antioxidants.

Please find the below comments and the attached annotated file. In addition, the researchers should respond to my comments, which are included in the attached file

The comments:

·  Please give numerical data or ratios including the impact of Eos as important antioxidants, in the abstract section.

· It is preferable if you can add graphical abstract so that it is easier to understand and clearer.

· Keywords: “I suggest rephrasing these words because keywords should not contain abbreviations.

·  Please insert the current work problem and the object of conducting it in the form of simplified sentences at the end of the introduction part

·  The materials and methods part is well written and detailed, while some parts only need a linguistic revision

·   Although the results of this MS are long, the results are clearly presented.

·        Discussion: the discussion is well written but the authors have to pay attention to the following comments for improving the manuscript.

-  In the discussion parts, a brief summary of the obtained result must first be mentioned, and then that result should be discussed from the authors’ point of view, with the help of previous articles in this concern.

- In the discussion section, conjunctions should be used to show the relationship between sentences.

-  The results and discussion must be merged as I mentioned in the comment (in the attached file)

- Conclusions: This is not a conclusion. The conclusion part should contain the most important results and recommendations of the current study in the form of a brief paragraph and simple and clear sentences, and not a repetition of what was mentioned more than once in the abstract and in the introduction as well.

·  References: Cross-check all the references for mistakes, especially the scientific names.

·        General comment:

·        Pls, shorten the long sentences, add recent ref. as soon as possible, and avoid repetition.

·  The manuscript contains some typo errors; please revise it carefully.

·  The usage of abbreviation should be used after the full term. Please be consistent with the usage of all abbreviations. Pls revise the abbreviations in the whole part of MS.

Round 2

Reviewer 3 Report

Reviewer comments-R2

Journal: Antioxidants (ISSN 2076-3921)

Manuscript IDantioxidants-2279242

Manuscript Title: “Essential oil composition, antioxidant activity and leaf micromorphology of five Tunisian Eucalyptus species”

After reviewing the modified manuscript carefully, I found that this manuscript has been improved after the authors responded to a number of my previous comments on it, but not all comments were responded to. Hence, there are some minor comments that need to be taken care before accepting for publishing. The following are some of the comments that may need to be made before publication:

Please find the below comments. In addition, the authors should respond to my comments.

The comments:

·  It is preferable if you can add graphical abstract so that it is easier to understand and clearer.

·  The materials and methods part is well written and detailed, while some parts need a linguistic revision.

·  In the discussion section, conjunctions should be used to show the relationship between sentences.

·  The results and discussion must be merged for improving the quality and clarity of the manuscript and prevent unjustified repetition

· Shorten the long sentences, add recent ref. as soon as possible, and avoid repetition.

·  There are some typo errors; please revise it carefully.

· The abbreviation should be used after the full term. Please be consistent with the usage of all abbreviations. Pls revise the abbreviations in the entire part of MS.

Author Response

Best regards

Laura Cornara
